# Ecological Environment Quality Assessment of Arid Areas Based on Improved Remote Sensing Ecological Index—A Case Study of the Loess Plateau

**Ming Shi** [1,2,3]**, Fei Lin** [1,3] **, Xia Jing** [2]**, Bingyu Li** [2]**, Yang Shi** [1,3] **and Yimin Hu** [1,3,*]

1    Institute of Intelligent Machines, Hefei Institutes of Physical Science, Chinese Academy of Sciences, Hefei 230031, China; mshi@iim.ac.cn (M.S.); feilin@iim.ac.cn (F.L.); shiyang@iim.ac.cn (Y.S.)
2    College of Geomatics, Xi'an University of Science and Technology, Xi'an 710054, China; jingxiaxust@163.com (X.J.); 22110010001@stu.xust.edu.cn (B.L.)
3    Hefei Institutes of Collaborative Research and Innovation for Intelligent Agriculture, Hefei 231131, China
*    Correspondence: ymhu@iim.ac.cn

**Abstract:** Ecosystems in arid and semi-arid areas are delicate and prone to different erosive effects. Monitoring and evaluating the environmental ecological condition in such areas contribute to the governance and restoration of the ecosystem. Remote sensing ecological indices (RSEIs) are widely used as a method for environmental monitoring and have been extensively applied in various regions. This study selects the arid and semi-arid Loess Plateau as the research area, in response to existing research on ecological monitoring that predominantly uses vegetation indices as monitoring indicators for greenness factors. A fluorescence remote sensing ecological index (SRSEI) is constructed by using monthly synthesized sun-induced chlorophyll fluorescence data during the vegetation growth period as a new component for greenness and combining it with MODIS product data. The study generates the RSEI and SRSEI for the research area spanning from 2001 to 2021. The study compares and analyzes the differences between the two indices and explores the evolution patterns of the ecosystem quality in the Loess Plateau over a 21-year period. The results indicate consistent and positively correlated linear fitting trend changes in the RSEI and SRSEI for the research area between 2001 and 2021. The newly constructed ecological index exhibits a higher correlation with rainfall data, and it shows a more significant decrease in magnitude during drought occurrences, indicating a faster and stronger response of the new index to drought in the research area. The largest proportions are found in the research area's regions with both substantial and minor improvements, pointing to an upward tendency in the Loess Plateau's ecosystem development. The newly constructed environmental index can effectively evaluate the quality of the ecosystem in the research area.

**Keywords:** RSEI; Loess Plateau; ecological environment; GOSIF

## 1. Introduction

Ecosystems worldwide are more vulnerable to human activities and resulting environmental changes than ever before [1,2]. In previous years, with the accelerated process of urbanization in China, different forms of land use have changed significantly, and the negative effects of human activity on the environment are now more obvious than ever. This has led to a variety of ecological environmental issues, such as soil erosion, soil degradation, and soil heavy metal pollution [3–7]. To address the increasingly severe ecological environmental problems, the Chinese Ministry of Environment put forward the ecological index (EI) in 2006. This index weighs the assessment markers of biological abundance, vegetation coverage, water network coverage, stress on the land, and pollution load to determine the ecological environment's quality [8]. This technique has been frequently used to evaluate ecological quality [9–11]. A major obstacle to the broad use of this strategy is the difficulty in gathering multiple indicators in practical applications, which results

in a protracted update cycle for the data. Remote sensing technology has the advantages of large area coverage and multi-scale capabilities, which enables rapid perception and acquisition of radiometric information from different components of the ecosystem, such as soil, lake water areas, vegetation, and buildings. Therefore, it is widely used in ecosystem surveys and assessments [12–15]. Similar research and applications are being conducted not only in China but also in the United States and other parts of the world [16–20].

Currently, the single index technique and the comprehensive index method are the two main types of methodologies for assessing ecological quality utilizing remote sensing technology. The single index technique keeps track of and assesses certain ecosystem components using data from remote sensing. For example, the vegetation index [21], the leaf area index [22], and net primary productivity [23] are used to monitor the vegetation coverage in the research area; urban heat island impact is assessed using land surface temperature [24]; and land surface moisture [25] and various drought indices [26] are used to assess drought conditions. However, the composition of ecosystems is diverse and complex, and the various components interact with each other. These methods cannot comprehensively evaluate the ecological environment changes in the research area. The comprehensive index method uses multiple monitoring factors to comprehensively evaluate the ecological environment quality. The most representative is a completely remote-sensing-based ecological environment evaluation method proposed by Xu Hanqiu [27]. This method selects heat, greenness, dryness, and moisture as monitoring indicators closely related to environmental quality. By integrating the monitoring effects of each indicator, a thorough and scientific evaluation of the ecological environment quality in the research zone is carried out. Therefore, it has been widely applied in environmental monitoring and evaluation [28–30]. Karbalaei Saleh Sajjad et al. [31] used the RSEI to assess the temporal and spatial changes in the ecological quality of Isfahan, a large city in Iran. The results showed that the ecological quality of the city did not maintain a stable trend during the study period, but fluctuated multiple times, which may be attributed to natural and human-induced changes during the study period. The RSEI was utilized by Chen Zhiyun et al. to analyze the environmental quality of the Meizhou region, and the results revealed an increase in high-quality ecological environment regions within the study area and the suitability of MODIS data for regional ecological environment quality evaluation [32]. Min An et al. [33] used MODIS data from the Three Gorges Ecological Economic Corridor from 2001 to 2019 to evaluate the ecological environment quality of the research area and analyze the impact of human activities on the local ecological environment. In addition, there have been some improvements to the RSEI [34–36]. Zhang Jing, Yang Liping, and others [37] incorporated aerosol optical thickness data (AOD) into the RSEI, proposed an improved remote sensing ecological index, and used this index to evaluate the ecological environment quality of Xi'an City. The results showed that the newly constructed index considers atmospheric pollution factors and contributes to a more comprehensive evaluation of the ecological environment quality of the research area. Jia Haowei et al. [38] evaluated the ecological environment of the Qaidam Basin using the modified remote sensing ecological index, and they explored its motivating reasons. Na Chen et al. [39] used the improved remote sensing ecological index (IRSEI) to effectively assess the urban ecological environment of Jining City. According to the findings, there are considerable seasonal variations in the ecological quality of the study region, and during the last 20 years, Jining City's ecological environment quality has improved generally.

The reflectance data provide the foundation of the majority of the remote sensing ecological indices or modified remote sensing ecological indices employed in the aforementioned investigations. Previous research has shown that common vegetation indices are not strong enough to reflect the photosynthetic capacity of vegetation when used as indicators of greenness, and they have obvious lag in response to vegetation stress such as drought [40–43]. In the last ten years, a new vegetation remote sensing technology known as solar-induced chlorophyll fluorescence (SIF) has emerged that can make up for the shortcomings of conventional optical remote sensing observations based on "greenness"

and offer a fresh approach for tracking extensive vegetation photosynthesis [44–47]. In addition, it has been confirmed that SIF responds faster and with a more significant decline to drought stress on a temporal scale. On a spatial scale, it responds differently to different gradients of drought stress, especially capturing mild drought stress more accurately and depicting a larger range of spatial losses [48,49]. In terms of data acquisition, a large amount of data support is required for large-scale ecological environment assessment, resulting in a geometric growth in the complexity and time consumption of data preprocessing. In order to address this issue, a planetary-scale geospatial analysis using cloud systems like Google Earth Engine (GEE) [50] is needed. It can provide rich open-source data and powerful computing services, greatly shortening the time for remote sensing data collection and processing.

Environmental systems in arid and semi-arid areas are delicate and quickly impacted by numerous erosion processes. Since the Loess Plateau is a typical dry and semi-arid ecological region in China, it is important to quickly and thoroughly examine the quality of the region's ecological environment since this information will be crucial for future research on water and soil conservation and environmental management in the area. Therefore, this study takes the Loess Plateau as the study area and utilizes the Google Earth Engine (GEE) platform to acquire remote sensing data of the study area. By incorporating fluorescence data into the construction of remote sensing ecological indices and using principal component analysis, a dataset of fluorescence remote sensing ecological indices from 2001 to 2021 in the study area was generated. In order to provide a theoretical foundation and method references for monitoring, assessing, and protecting the ecological environment quality in the Loess Plateau region, the spatiotemporal variation patterns of ecological environment quality in the Loess Plateau from 2001 to 2021 were examined.

## 2. Experimental Materials and Methods

### 2.1. Introduction to the Study Area

The Loess Plateau, which extends over the administrative areas of Shaanxi, Shanxi, Ningxia, Qinghai, Gansu, Henan, and Inner Mongolia, is located in Central China's northern region. It is one of China's four main plateaus. The location of the study area and the various types of land use are shown in Figure 1. The Loess Plateau has a length of over 1000 km from east to west and a width of 750 km from north to south. It has an overall southeast low and northwest high topography, with an elevation decreasing in a wave-like pattern from northwest to southeast. The elevations in the study area range from 75 to 5149 m. The average annual temperature is between 9 and 12 degrees Celsius, with hot, wet summers and cold, dry winters. The climate is classified as a dry continental monsoon climate. The research area is situated in an arid and semi-arid environment, with precipitation averaging 100–800 mm per year on average and decreasing steadily from southeast to northwest. The study area is mainly characterized by the Yellow River as the main water system, with major tributaries including the Wei River and Fen River. The soil in the Loess Plateau is mainly loess, which has poor forest resources. It is a typical barren area with low vegetation coverage and poor water storage capacity, making it susceptible to various erosion processes and drought stress. The Loess Plateau's ecological system is severely vulnerable. It is susceptible to ecological issues as a result of the combined effects of climate change and human disruptions, which are mostly manifested as frequent natural catastrophes, water resource shortages, severe desertification, and severe soil erosion.

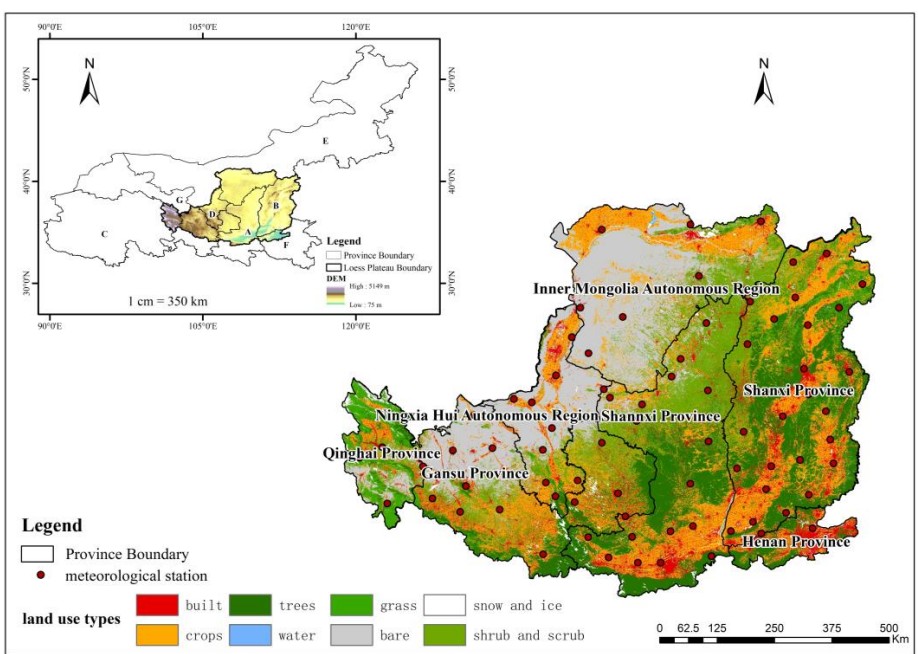

**Figure 1.** Study area location and land use map. A, B, C, D, E, F, and G stand in for Shaanxi Province, Shanxi Province, Qinghai Province, the Ningxia Hui Autonomous Region, the Inner Mongolia Autonomous Region, Henan Province, and Gansu Province, respectively.

### 2.2. Data Acquisition and Preprocessing

In this paper, SIF data from May to September 2001 and MODIS product data from May to September 2001 from 2001 to 2021 were selected based on the GEE platform as the main data sources (Table 1).

**Table 1.** Data sources and description.

| Satellite Data | Parameter | Temporal Resolution | Spatial Resolution |
|---|---|---|---|
| MOD09A1 | Reflectance products | 8 day | 500 m |
| MOD11A2 | Surface temperature | 8 day | 500 m |
| MOD13A1 | Vegetation indices | 16 day | 500 m |
| GOSIF | SIF | 30 day | 500 m |

The above MODIS product data were sourced from the GEE platform database. The fluorescence data were obtained from the Li and Xiao-produced worldwide monthly syntheses dataset based on OCO-2, which covered the years 2001 to 2021 [51]. The MODIS data were directly accessed online, and their spatial resolution was resampled to 500 m. To lessen the influence of water bodies on the humidity index, the impact of water information was removed using an improved Normalized Difference Water Index (MNDWI) [52,53]. The cloud masking tool was applied to remove clouds from the images. Subsequently, the maximum value synthesis was performed on the fluorescence data from May to September each year, and their spatial resolution was also resampled to 500 m to match the MODIS data. Then, the maximum value synthesis was applied to the MODIS data and fluorescence data to obtain annual data, which serve as the preprocessing dataset for calculating RSEI and SRSEI in this study.

### 2.3. Remote Sensing Ecological Monitoring Indicators

Greenness, humidity, temperature, and dryness are important indicators that are closely related to human survival and reflect the quality of the environment. They also serve as indicators of the excellence or inferiority of the ecological environment, which people are capable of seeing instinctively. Therefore, these four aspects must be taken into

account while creating remote sensing ecological indices. Remote sensing pictures may be used to extract parameters relating to these four characteristics by using digital image thematic information processing techniques. Figure 2 displays the study's flowchart.

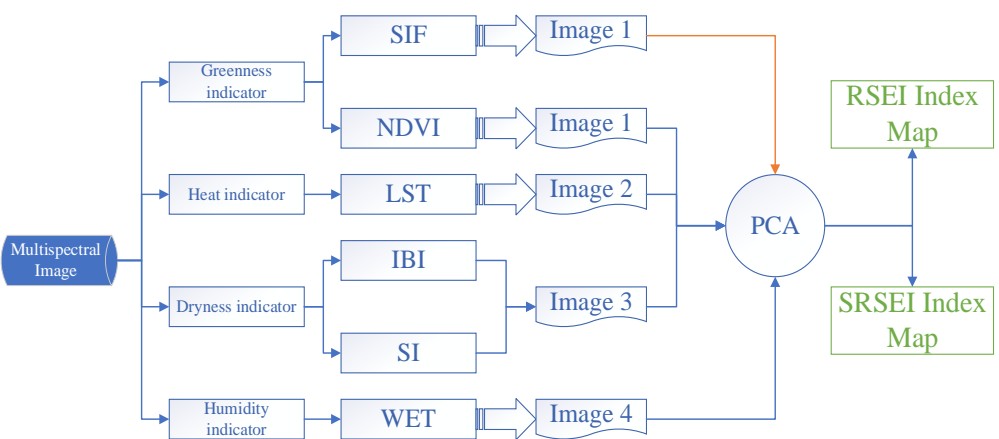

**Figure 2.** Flow chart of remote sensing ecological index construction.

First, MODIS surface reflectance data and surface temperature data were acquired through GEE. The MOD13A1 dataset was used to obtain the NDVI (Normalized Difference Vegetation Index, NDVI), while the MOD09A1 dataset provided surface humidity wetness (WET), the NDBI (Normalized Difference Building Index, NDBI), and the SI (Soil Index, SI). The MOD11A2 dataset was used to obtain land surface temperature (land surface temperature, LST). The NDVI was used as the greenness indicator, LST as the temperature indicator, NDSI (Normalized Difference Soil Index, NDSI, derived from NDBI and SI) as the dryness indicator, and WET as the humidity indicator. The NDVI was chosen as the greenness indicator for RSEI calculation, and SIF was chosen as the greenness indicator for SRSEI calculation. These indicators were combined with temperature, humidity, and dryness indicators. A principal component analysis (PCA) was employed to fuse the indicators and construct the RSEI and SRSEI for the study area.

The NDVI is one of the commonly used indicators to characterize vegetation coverage [54,55]. Green vegetation has a low reflectance in the red band and a high reflectivity in the near-infrared band because it absorbs incoming solar energy in the red band and scatters it in the near-infrared band. By linearly combining the red and near-infrared bands, the NDVI can be used as an indicator of vegetation growth status. In the construction of ecological indices, the NDVI is often used as the greenness indicator, as shown in Equation (1).

$$NDVI = \frac{NIR - R}{NIR + R} \tag{1}$$

where *NIR* represents the near-infrared band reflectance and *R* represents the red band reflectance.

In the calculation of the humidity index, this study uses the MODIS reflectance dataset MOD09A1 and obtains the surface humidity component through Tasseled Cap Transformation (TCP). TCP is a linear transformation proposed by Kauth et al. [56] that can eliminate spectral response correlation. The third component of this transformation represents the humidity index, which reflects the humidity status of vegetation, soil, and water bodies. TCP is commonly used in Landsat and IKONOS images, but recent studies have found its presence in MODIS as well [57]. This work used the MOD09A1 surface reflectance product to compute the WET index in the study region based on the enhanced MODIS Tasseled Cap Transformation method [58], as given in Equation (2).

$$WET = 0.1147\,R + 0.2489\,NIR_1 + 0.2408\,B + 0.3132\,G - 0.3122\,NIR_2 - 0.6416\,SWIR_1 - 0.5087\,SWIR_2 \tag{2}$$

In the equation, $NIR_1$ represents near-infrared band 1, $B$ represents the blue band, $G$ represents the green band, $NIR_2$ represents near-infrared band 2, $SWIR_1$ represents shortwave infrared band 1, and $SWIR_2$ represents shortwave infrared band 2.

The Normalized Difference Building Index (NDBI) [59] and the soil index (SI) [60] were utilized as composite indicators of dryness in the research region since bare soil and urban development land play a significant part in the cause of the study area's dryness. The calculation of the NDBI is shown in Equation (3). The surface reflectance data used for this calculation are the same as those used for calculating the humidity index.

$$NDBI = \frac{2SWIR_1/(SWIR_1 + NIR) - [NIR/(NIR + R) + G/(G + SWIR_1)]}{2SWIR_1/(SWIR_1 + NIR) + [NIR/(NIR + R) + G/(G + SWIR_1)]} \quad (3)$$

In the equation, $R$ represents the red band, $NIR$ represents the near-infrared band, $G$ represents the green band, $SWIR_1$ represents shortwave infrared band 1, and $SWIR_2$ represents the shortwave infrared band.

The bare soil index (SI) calculation is shown in Equation (4):

$$SI = \frac{(SWIR_1 + R) - (NIR + B)}{(SWIR_1 + R) + (NIR + B)} \quad (4)$$

In the equation, $R$ represents the red band, $NIR$ represents the near-infrared band, $B$ represents the blue band, and $SWIR_1$ represents shortwave infrared band 1.

The dryness index (NDSI) is shown in Equation (5):

$$NDSI = \frac{SI + NDBI}{2} \quad (5)$$

In the equation, NDSI represents the dryness index and SI and NDBI represent the bare soil index and Normalized Building Index, respectively.

Surface temperature can be used to represent the distribution of heat in the study area and was therefore chosen as the thermal index. Utilizing the MOD11A2 surface temperature dataset and applying it to Equation (6), it is converted into Celsius to obtain the thermal index in the study area.

$$LST = LST_{MOD11A2}0.02 - 273.15 \quad (6)$$

In the equation, $LST_{MOD11A2}$ represents the original data from the dataset, and $LST$ represents the converted land surface temperature data.

For the processing of monthly GOSIF data, the maximum value synthesis, projection, and resampling operations are performed as preprocessing steps. Then, the data are clipped using the vector boundary of the study area to obtain the fluorescence dataset for the period from 2001 to 2021.

### 2.4. Construction of SRSEI

Due to the inconsistent units and ranges of different factors, it is necessary to normalize each factor separately before conducting a principal component analysis (PCA). The normalization method was applied to each factor using the formula shown in Equation (7):

$$Fa_i = \frac{I_i - I_{\min}}{I_{\max} - I_{\min}} \quad (7)$$

In the equation, $Fa_i$ represents the normalized value of a specific factor, $I_i$ represents the original value of the factor, and $I_{\max}$ and $I_{\min}$ represent the maximum and minimum values of that factor, respectively.

Based on the PCA method, the RSEI values and four factors were calculated for the study area from 2001 to 2021. In the ArcGIS platform, band synthesis was performed on

each factor, and the fluorescence data were used to replace the original NDVI data as a new greenness indicator. Subsequently, a PCA was conducted on the synthesized factors.

$$SRSEI_0 = PC_1[f(GOSIF, WET, NDSI, LST)] \tag{8}$$

Among these, $SRSEI_0$ denotes the first PCA-derived fluorescence remote sensing ecological index. $PC_1$ represents the first principal component of the PCA; f denotes the normalization process applied to each factor. When constructing the RSEI and SRSEI using a PCA, only when the greenness and humidity indicators are negative and the temperature and aridity indicators are positive, the value of 1-$PC_1$ is used.

After normalizing the obtained $RSEI_0$ and $SRSEI_0$, the RSEI and SRSEI values for the study area were obtained. The RSEI and SRSEI were classified into five categories based on the categorization standards for ecological habitats found in pertinent literature [61–64]: poor (0–0.2), fair (0.2–0.4), moderate (0.4–0.6), good (0.6–0.8), and excellent (0.8–1).

### 3. Results and Analysis

#### 3.1. Analysis by Principal Component of the SRSEI Indicators

A statistical analysis was conducted on the results of the PCA for the study area from 2001 to 2021 (Figure 3). From the graph, it can be observed that except for the year 2003, the first principal component of the greenness and humidity indices for all other years is positive, while the dryness and thermal indices are negative. Therefore, when calculating the SRSEI, PC1 was used for all years except 2003, where 1-PC1 was used. Between 2001 and 2021, the study area's contribution rate of the PC1 primary component to the SRSEI is larger than 67%, with a range of 87% to 67.65%. This indicates that PC1 effectively captures the characteristic information of various indices, justifying the use of PC1 to construct the SRSEI for evaluating the ecological environment quality in the Loess Plateau.

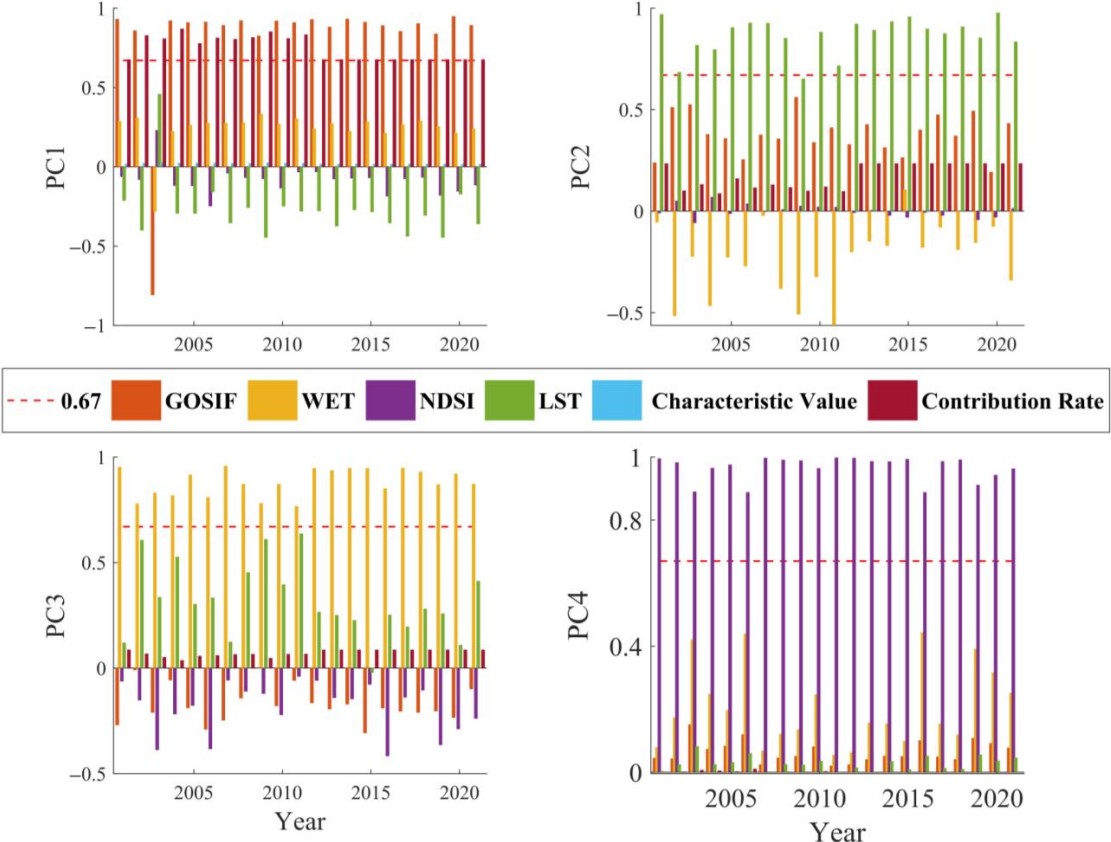

**Figure 3.** Principal component analysis of SRSEI.

### 3.2. RSEI vs. SRSEI in the Loess Plateau

An analysis was conducted on the RSEI and SRSEI anomalies (left) and means (right) in the research region from 2001 to 2021 (Figure 4). According to earlier study findings [65–69], both the RSEI and SRSEI for the anomalies show consistent trend changes, confirming the positive association and pointing to an increase in the ecological environmental quality in the Loess Plateau region. As for the RSEI, the range of anomaly values is between −0.11 and 0.06. Among them, for 9 years, the anomaly values are negative, and the maximum absolute value of the anomalies occurred in 2004, reaching 0.114. As for the SRSEI, the range of anomaly values is between −0.08 and 0.06. Among them, for 13 years, the anomaly values are negative, and the maximum absolute value of the anomalies occurred in 2001, reaching 0.08. In terms of means, the overall mean of the SRSEI is lower than that of the RSEI, which may be due to the influence of vegetation phenology and drought stress in the study area, leading to a significant decrease in the greenness index GOSIF used in the SRSEI compared to the NDVI. The range of RSEI values is [0.36, 0.53], with a mean of 0.47. The year with the highest RSEI value is 2019, and the year with the lowest RSEI value is 2004. The range of SRSEI values is [0.29, 0.43], with a mean of 0.37. The year with the highest SRSEI value is 2018, and the year with the lowest SRSEI value is 2001.

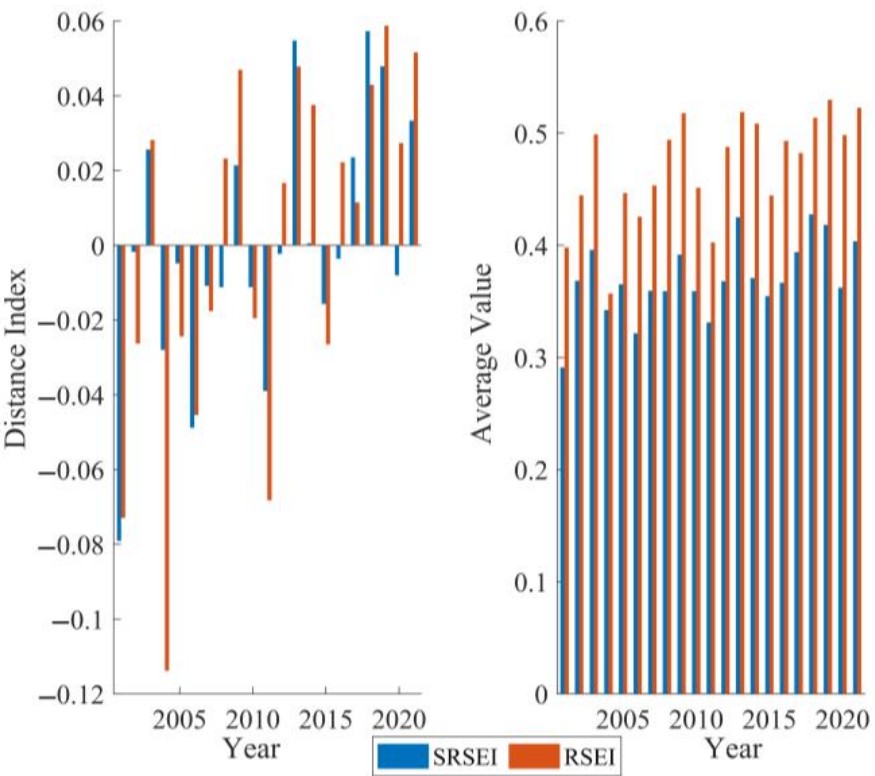

**Figure 4.** RSEI and SRSEI anomaly analysis and mean values from 2001 to 2021.

The difference between the mean values of the RSEI and SRSEI for each year is calculated and presented in Table 2. The largest difference occurs in 2014, with a value of 0.138, while the smallest difference occurs in 2004, with a value of 0.015. It can be seen that there are only minor mean changes between the RSEI and SRSEI.

**Table 2.** Comparison between RSEI and SESEI.

| Year | 2001 | 2002 | 2003 | 2004 | 2005 | 2006 | 2007 | 2008 | 2009 | 2010 | 2011 |
|------|------|------|------|------|------|------|------|------|------|------|------|
| Difference | 0.107 | 0.076 | 0.103 | 0.015 | 0.081 | 0.104 | 0.094 | 0.135 | 0.126 | 0.092 | 0.071 |
| **Year** | **2012** | **2013** | **2014** | **2015** | **2016** | **2017** | **2018** | **2019** | **2020** | **2021** | |
| Difference | 0.120 | 0.094 | 0.138 | 0.090 | 0.127 | 0.089 | 0.086 | 0.112 | 0.136 | 0.119 | |

The China Meteorological Administration (CMA) provided the monthly precipitation data for 77 meteorological stations in the research region from 2001 to 2018. Using a time scale of 5, the cumulative precipitation for the study area was calculated. Subsequently, the RSEI and SRSEI values for each station were extracted and analyzed for their correlation with the cumulative precipitation. Table 3 presents the outcomes.

**Table 3.** Correlation between RSEI and SRSEI and accumulated precipitation at 77 meteorological stations during 2001–2018.

| Index/Year | 2001 | 2002 | 2003 | 2004 | 2005 | 2006 | 2007 | 2008 | 2009 |
|------------|------|------|------|------|------|------|------|------|------|
| RSEI | 0.05 | 0.24 | 0.55 | 0.4 | 0.58 | 0.44 | 0.39 | 0.27 | 0.59 |
| SRSEI | 0.16 | 0.2 | 0.56 | 0.51 | 0.65 | 0.54 | 0.51 | 0.39 | 0.58 |
| **Index/Year** | **2010** | **2011** | **2012** | **2013** | **2014** | **2015** | **2016** | **2017** | **2018** |
| RSEI | 0.42 | 0.49 | 0.42 | 0.51 | 0.39 | 0.39 | 0.33 | 0.49 | 0.46 |
| SRSEI | 0.56 | 0.53 | 0.35 | 0.44 | 0.45 | 0.52 | 0.34 | 0.43 | 0.47 |

From Table 3, the SRSEI, RSEI, and cumulative precipitation from May to September in the research region from 2001 to 2018 may be found to be significantly correlated. Additionally, the correlation coefficient between the SRSEI and cumulative precipitation is larger than that between the RSEI and cumulative precipitation. This indicates that the SRSEI is more susceptible to precipitation information and indirectly reflects the drought condition in the study area. Among them, the year with the highest correlation between the RSEI and cumulative precipitation occurred in 2009, with a value of 0.59. The average correlation over 18 years is 0.41. The year with the highest correlation between the SRSEI and cumulative precipitation occurred in 2005, with a value of 0.65. The average correlation over 18 years is 0.46.

Previous studies have shown [70–72] that compared to the NDVI, SIF can more accurately monitor the impact of drought events on vegetation. Therefore, in the construction of the SRSEI, fluorescence data that can reflect vegetation physiological activity and photosynthetic capacity were added, which can better reflect the vegetation activity and photosynthetic capacity in the study area and exhibit a more sensitive and rapid response to drought and other stresses.

To investigate the relationship between the constructed ecological environmental monitoring index and its component factors, a correlation analysis was conducted between the RSEI and SRSEI and the corresponding greenness index GOSIF, NDVI, heat index LST, dryness index NDSI, and wetness index WET in the study area (Figure 5). From the figure, it can be observed that both the RSEI and SRSEI have a close and significant or highly significant correlation with each index. This shows that the study area's ecological environmental quality can be captured by both the SRSEI and the RSEI. Furthermore, it demonstrates that in the construction of the index, the RSEI or the SRSEI fully utilizes the information of dryness, wetness, greenness, and heat in the study area, making the evaluation of ecological environmental quality more scientifically reasonable compared to using only a single piece of information.

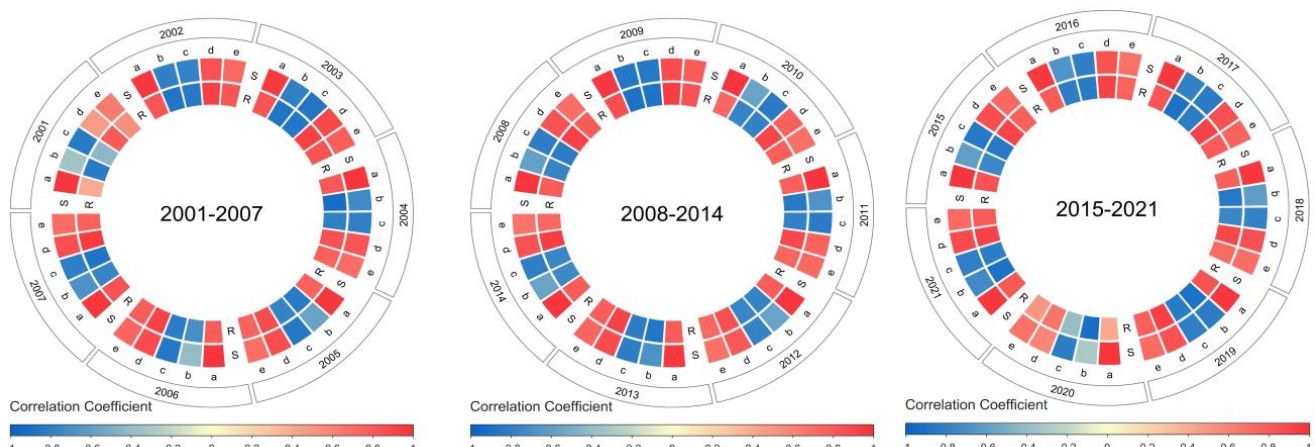

**Figure 5.** Correlation of indicators in the Loess Plateau during 2001–2021.

In the Figure 5, S and R represent the SRSEI and RSEI, respectively, while a, b, c, d, and e represent the GOSIF, LST, NDSI, NDVI, and WET, respectively.

### 3.3. RSEI and SRSEI's Geographic Distribution in the Loess Plateau Area

The RSEI and SRSEI were reclassified, and the means of different provinces were calculated. The RSEI and SRSEI values in the various provinces of the research region from 2001 to 2021 are depicted spatially in Figure 6. The left plot represents the annual RSEI, while the right plot represents the annual SRSEI. Overall, the environmental quality in the Loess Plateau region has improved. The ecological environmental quality of the Loess Plateau region exhibits considerable spatial heterogeneity features, demonstrating a pattern of poor in the northwest and high in the southeast, as can also be shown in Figure 6. While regions with low RSEI and SRSEI values are mostly found in the north and northwest of the Inner Mongolia Autonomous Region, the Ningxia Hui Autonomous Region, and Gansu Province, regions with high RSEI and SRSEI values are primarily found in Shaanxi, Shanxi, Henan, and Qinghai Provinces. This may be due to the presence of large areas with vegetation coverage, such as deserts, resulting in a decrease in greenness and wetness indicators that are positively correlated with ecological environmental indices, as well as an increase in dryness and heat indicators that are negatively correlated with ecological environmental indices, leading to a downward trend in the overall ecological environmental index.

For the period from 2001 to 2021, the regions of extremely poor, poor, moderate, good, and excellent levels were estimated in order to examine changes in the various ecological environment levels in the research area. The outcomes are displayed in Figure 7.

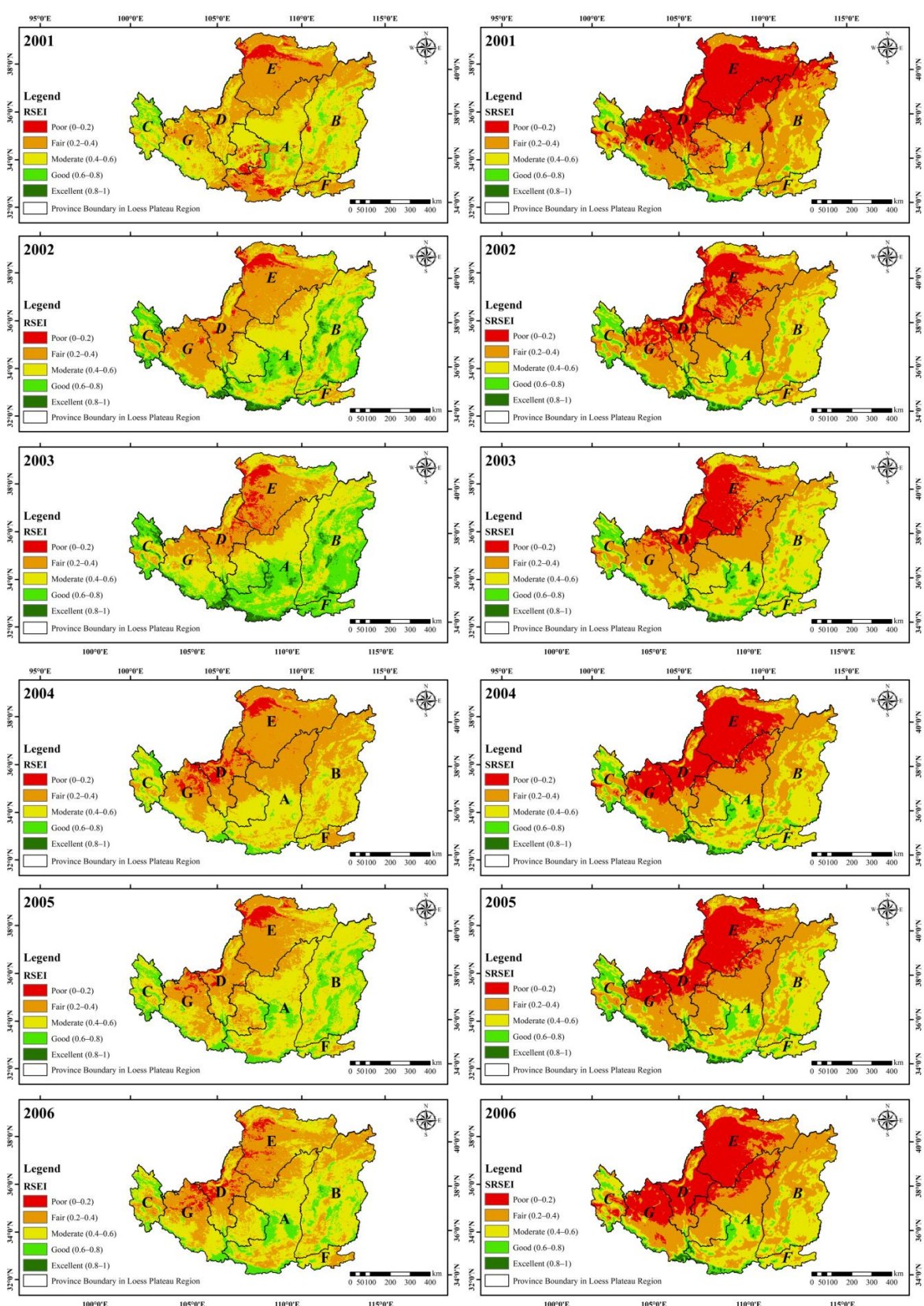

**Figure 6.** *Cont.*

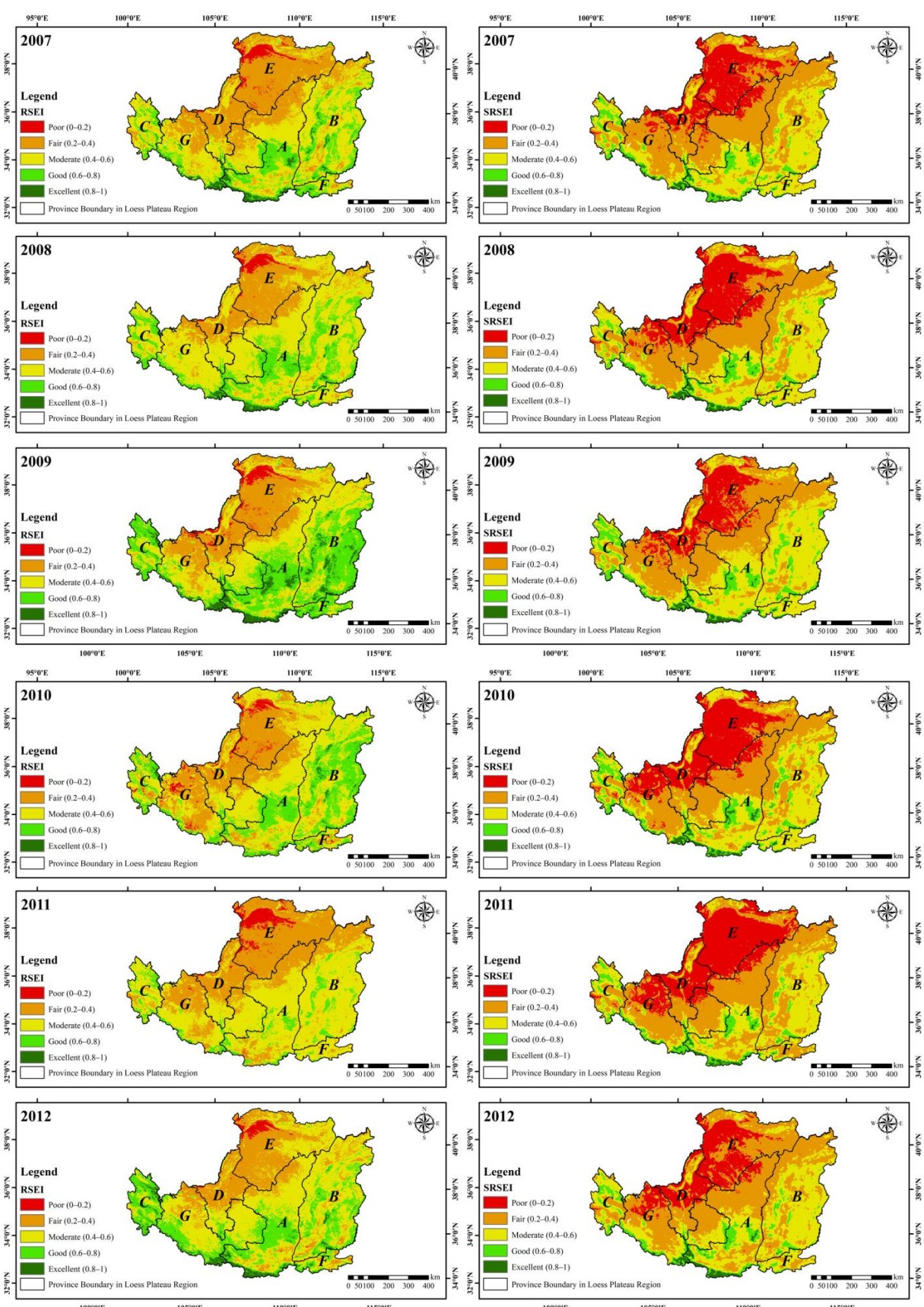

**Figure 6.** *Cont.*

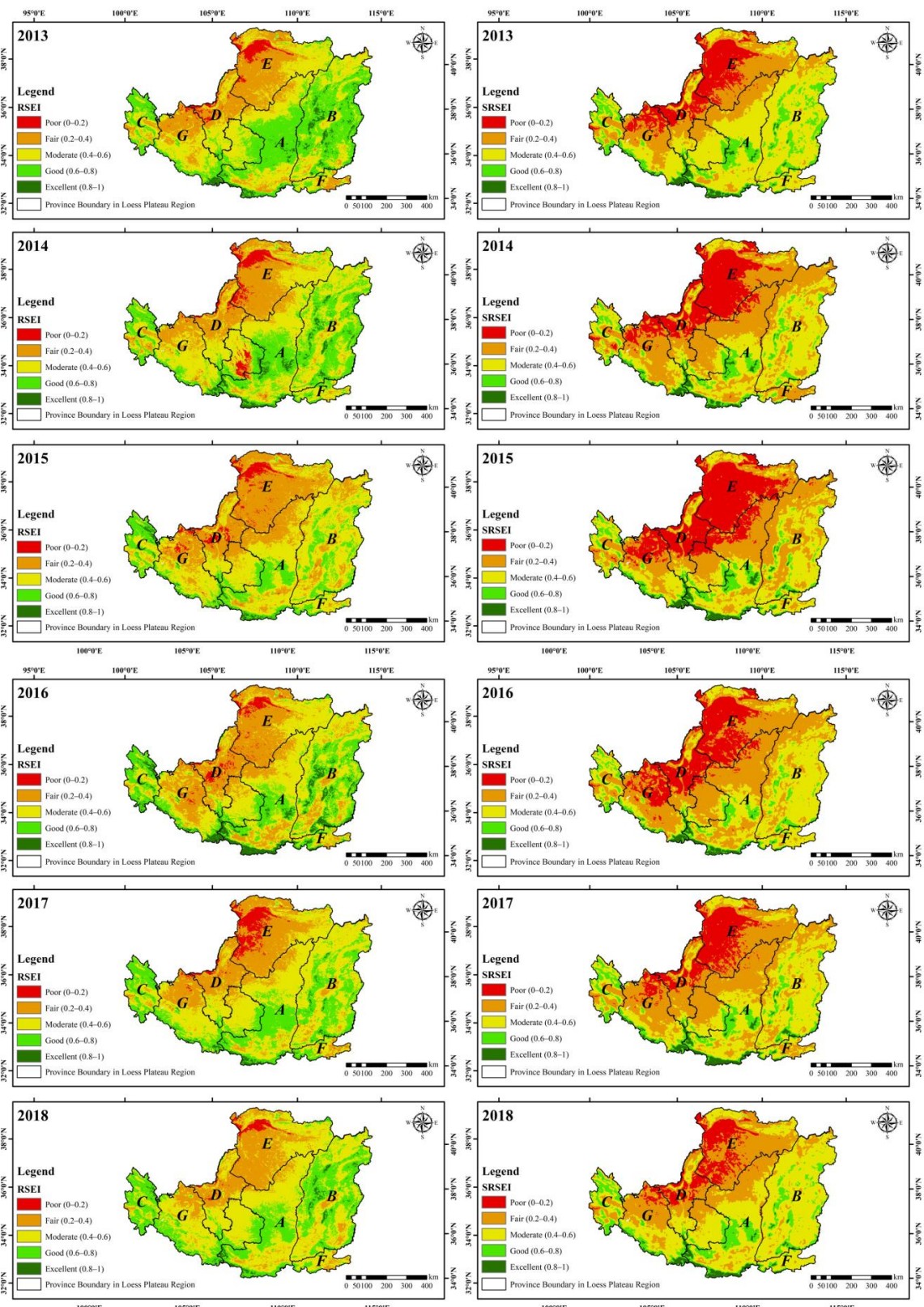

**Figure 6.** *Cont.*

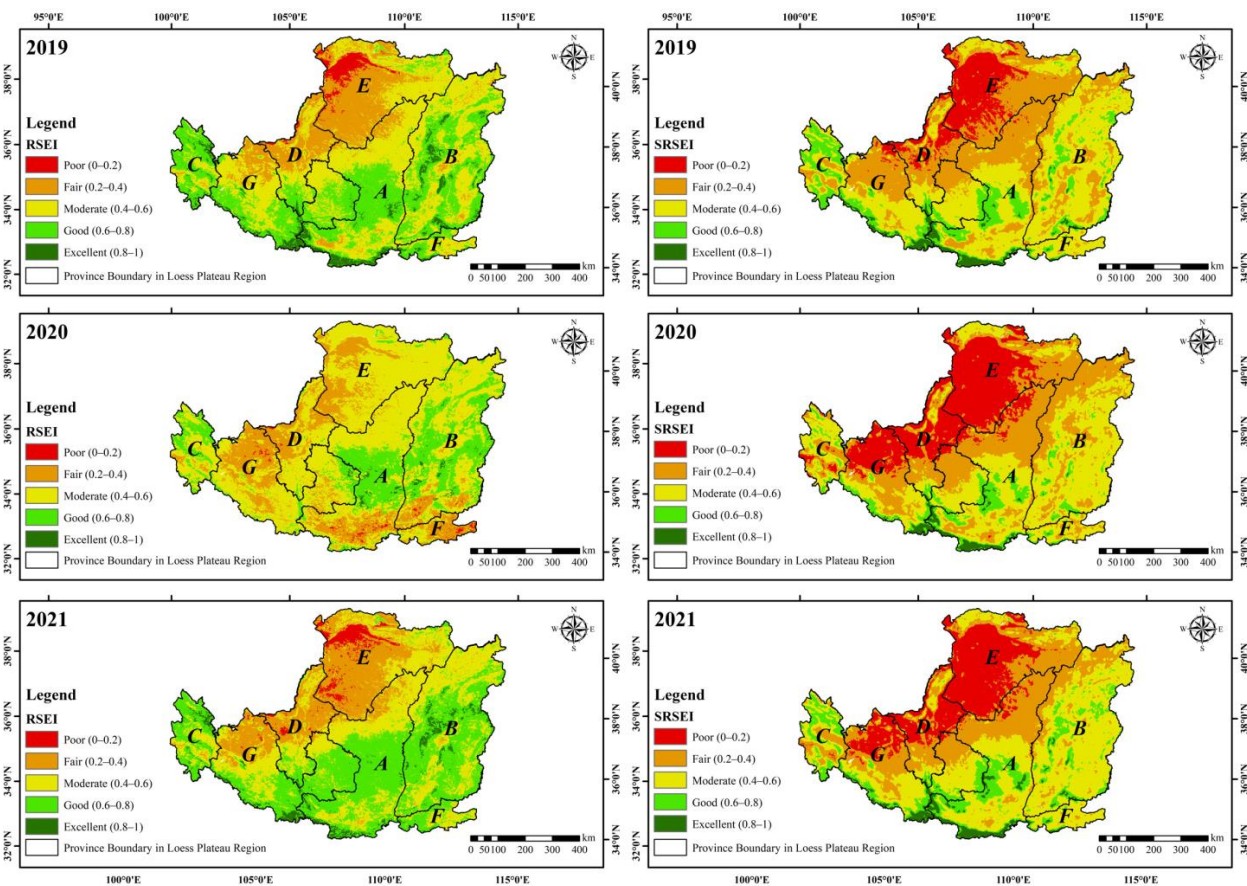

**Figure 6.** Distribution maps of RSEI levels in Loess Plateau from 2001 to 2021. A, B, C, D, E, F, and G stand in for Shaanxi Province, Shanxi Province, Qinghai Province, the Ningxia Hui Autonomous Region, the Inner Mongolia Autonomous Region, Henan Province, and Gansu Province, respectively.

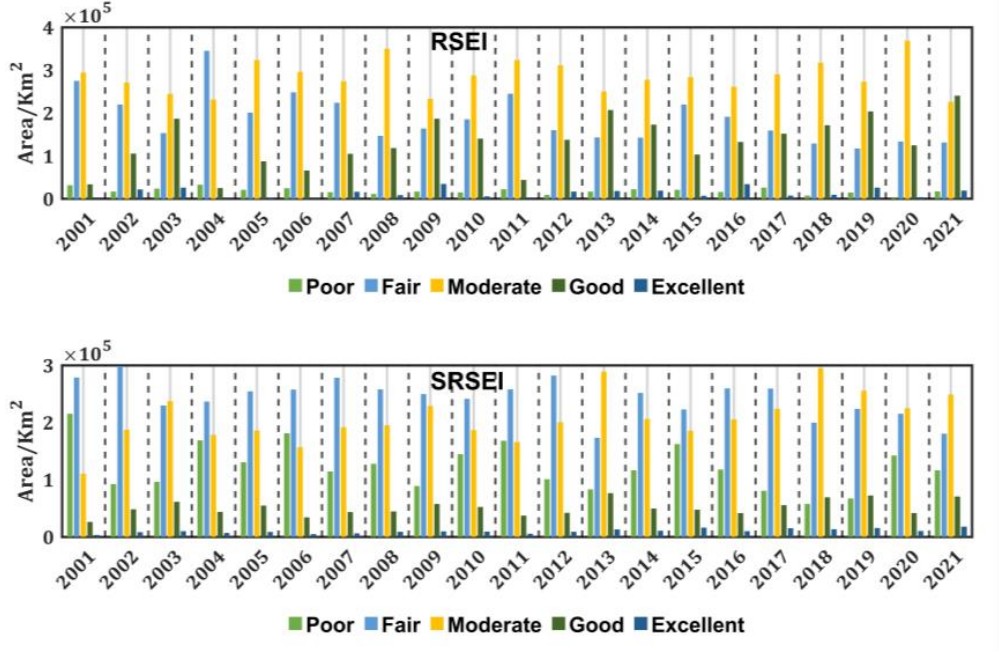

**Figure 7.** Area statistics of RSEI and SRSEI grades in the Loess Plateau during 2001–2021.

From Figure 6, it can be observed that both indices show a decreasing trend in the areas of poor and very poor levels and an increasing trend in the areas of moderate, good, and excellent levels in the study area from 2001 to 2021. Over the 21-year period, for the RSEI, compared to 2001, the areas of poor and very poor levels decreased by 0.51% in 2021, while the area of good and excellent levels increased by 6.6%. As for the SRSEI, the areas of poor and very poor levels decreased by 0.4%, and the areas of good and excellent levels increased by 1.96%. In terms of poor and very poor levels, the SRSEI detects more area than RSEI. Compared to the greenness index, the NDVI used in the RSEI, the use of the fluorescence index SIF in the SRSEI is more closely related to vegetation vitality and can better reflect the vegetation growth status in the study area. Additionally, when vegetation is under short-term stress, the SIF value decreases rapidly, while the NDVI value exhibits noticeable lag, resulting in the SRSEI being more sensitive than the RSEI in reflecting the impact of stress in the study area. This leads to a more objective evaluation of ecological environmental quality.

### 3.4. Comparison of RSEI and SRSEI Levels among Provinces in the Loess Plateau Region

The areas of various RSEI and SRSEI grades from 2001 to 2021 were tallied in order to examine the ecological environment quality of several provinces in the Loess Plateau region, as illustrated in Figure 8. Figure 8a presents the RSEI classification map of the provinces in the research area, while Figure 8b displays the SRSEI classification map of the provinces. A, B, C, D, E, F, and G stand in for Shaanxi Province, Shanxi Province, Qinghai Province, the Ningxia Hui Autonomous Region, the Inner Mongolia Autonomous Region, Henan Province, and Gansu Province, respectively. From Figure 8, it can be observed that for Shaanxi Province, Shanxi Province, Henan Province, and Qinghai Province, the proportion of areas classified as moderate, good, and excellent in both the RSEI and SRSEI is much greater than the proportion of areas classified as very poor and poor. The number of places categorized as extremely poor and poor, however, is disproportionately high for the Ningxia Hui Autonomous Region, the Inner Mongolia Autonomous Region, and Gansu Province, showing a worsening in local environmental quality.

From the above figure, it can be inferred that for the RSEI from 2001 to 2021, in Shaanxi Province, Shanxi Province, and Qinghai Province, the areas classified as very poor, poor, and moderate show a decreasing trend, while the areas classified as good and excellent show an increasing trend. In the Ningxia Hui Autonomous Region, the Inner Mongolia Autonomous Region, and Gansu Province, the areas classified as very poor and poor exhibit a decreasing trend, while the areas classified as moderate, good, and excellent show an increasing trend. In Henan Province, the area classified as poor shows a decreasing trend, while the areas classified as very poor, moderate, good, and excellent show an increasing trend.

The extremely poor and poor regions in Shaanxi Province, Shanxi Province, Qinghai Province, and Gansu Province exhibit a declining tendency for the SRSEI from 2001 to 2021, whereas the moderate, good, and excellent areas show a growing trend. In the Ningxia Hui Autonomous Region, the area classified as very poor shows a decreasing trend, while the areas classified as poor, moderate, good, and excellent show an increasing trend. In the Inner Mongolia Autonomous Region, the area classified as very poor shows a decreasing trend, while the areas classified as poor, moderate, and good show an increasing trend. In Henan Province, the areas classified as very poor and moderate show a decreasing trend, while the areas classified as poor, good, and excellent show an increasing trend.

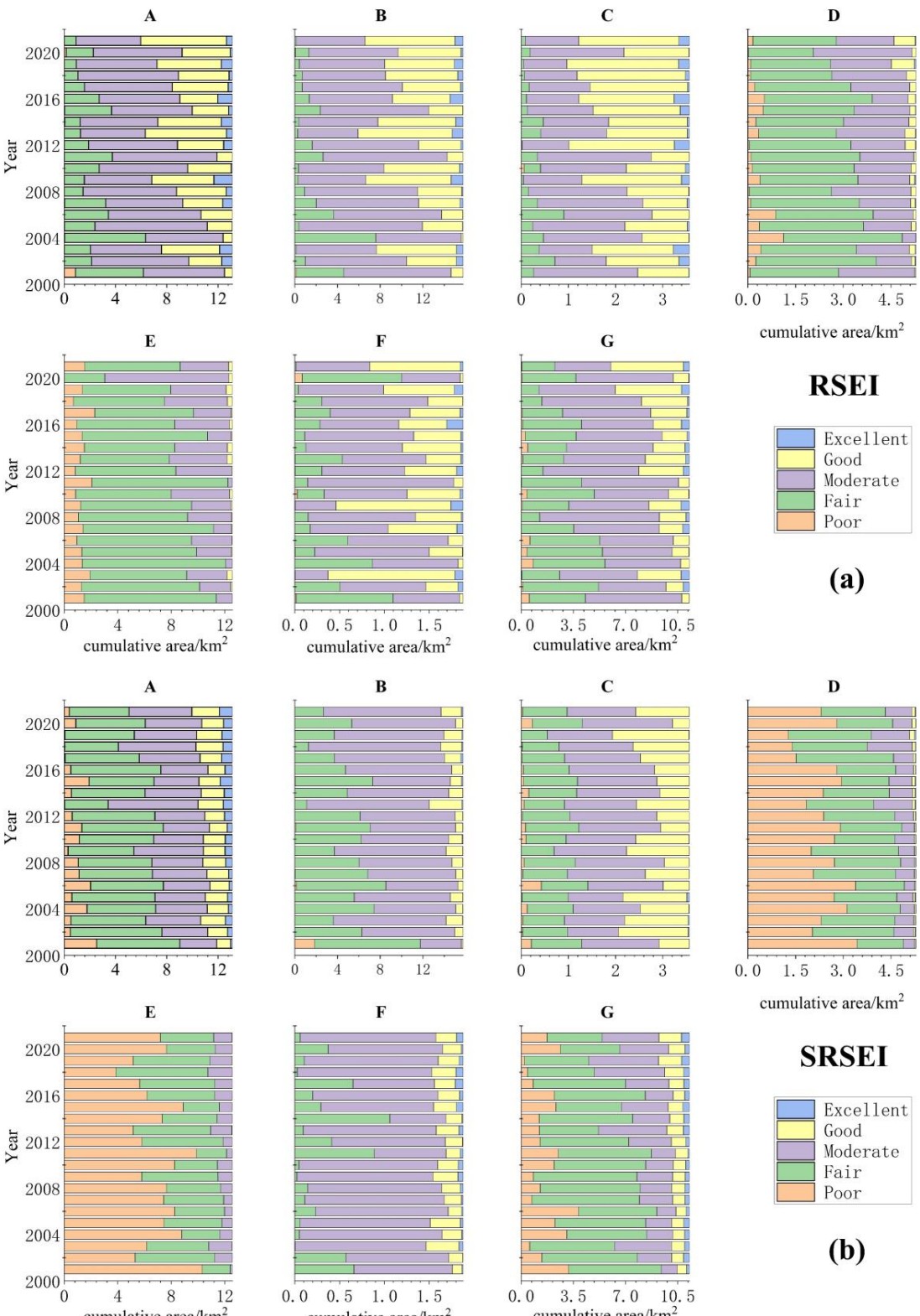

**Figure 8.** RSEI and SRSEI grading area ratio of provinces in Loess Plateau region from 2001 to 2021 (×10⁵ km²). (**a**) presents the RSEI classification map of the provinces in the research area. (**b**) displays the SRSEI classification map of the provinces. A, B, C, D, E, F, and G stand in for Shaanxi Province, Shanxi Province, Qinghai Province, the Ningxia Hui Autonomous Region, the Inner Mongolia Autonomous Region, Henan Province, and Gansu Province, respectively.

### 3.5. Trends of RSEI and SRSEI in the Loess Plateau

The Slope Trend Analysis method is a commonly used data trend analysis method that involves performing a linear regression analysis on a variable that changes over time to determine its trend in the time dimension [73]. Equation (9) displays the calculation formula for this strategy:

$$Slope = \frac{n \sum\limits_{i=1}^{n} (iSRSEI_i) - \sum\limits_{i=1}^{n} i \sum\limits_{i=1}^{n} SRSEI_i}{n \sum\limits_{i=1}^{n} i^2 - (\sum\limits_{i=1}^{n} i)^2} \tag{9}$$

In Equation (9), $n$ represents the duration of the study period, and $SRSEI_i$ represents the SRSEI image of the $i$-th year. When $Slope > 0$, it indicates that the pixel shows an increasing trend, and when $Slope < 0$, it indicates a decreasing trend. Then, the generated trend value images were subjected to the Mann–Kendall trend test for significance judgment. Taking a confidence level of 0.05, the study area can be classified into five categories: severe degradation, slight degradation, stable, slight improvement, and significant improvement (Figure 9).

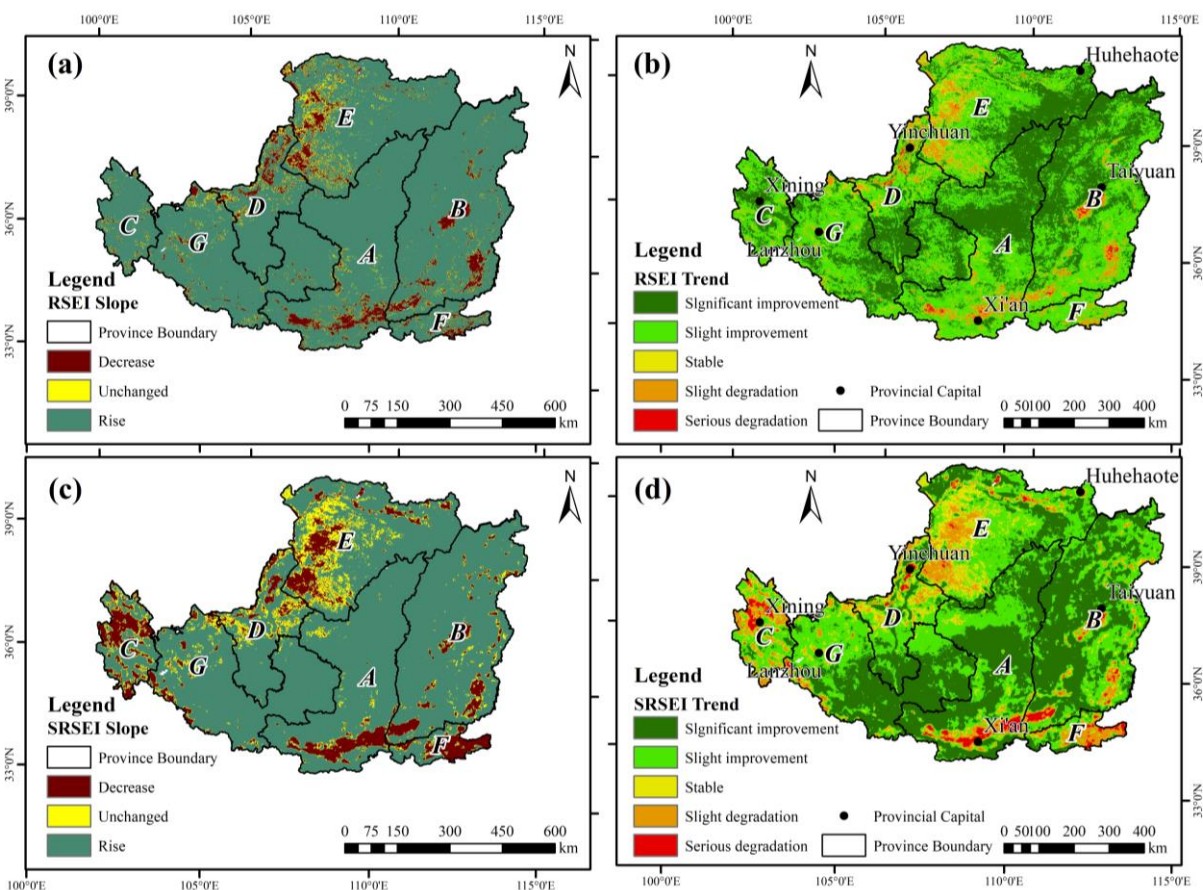

**Figure 9.** Trends of RSEI and SRSEI in various provinces in the Loess Plateau region from 2001 to 2021. (**a**,**c**) represent the slope analysis results of RSEI and SRSEI, respectively. (**b**,**d**) represent the Mann–Kendall trend analysis results of RSEI and SRSEI, respectively. A, B, C, D, E, F, and G stand in for Shaanxi Province, Shanxi Province, Qinghai Province, the Ningxia Hui Autonomous Region, the Inner Mongolia Autonomous Region, Henan Province, and Gansu Province, respectively.

The slope trend classification map and the Mann–Kendall trend analysis classification map of the RSEI in the study area are shown in Figures 9a and 9b respectively, while the

slope trend classification map and the Mann–Kendall trend analysis classification map of the SRSEI are shown in Figure 9c,d respectively.

As can be observed from Figure 9a, the RSEI exhibits an upward trend in the majority of the Loess Plateau locations. According to the analysis of each grade, the region with a declining trend makes up 7% of the total area, the area with a steady trend makes up 4.9% of the whole area, and the area with a growing trend makes up 88.1% of the entire area. The research area's primary change pattern may be shown to be the growing trend. It is evident that the research area's predominant changing pattern is an upward trend. Figure 9b shows that the majority of regions in the Loess Plateau indicate an improving tendency, with the middle portion of the plateau housing the majority of these places. Analyzing each grade, the area with a significant improvement is 243,014.25 km$^2$, accounting for 38.3% of the total area; the area with slight improvement is 316,108.5 km$^2$, accounting for 49.8% of the total area; and the stable area is 30,951.5 km$^2$, accounting for 4.9% of the total area. The highly damaged area is 2621 km$^2$, making up 0.4% of the overall area, while the mildly degraded area is 42,007.75 km$^2$, making up 6.6% of it. The dominant trends are significant improvement and slight improvement, indicating an improvement in the environmental quality of the Loess Plateau.

Figure 9c demonstrates that the SRSEI exhibits an upward trend in the majority of Loess Plateau locations. The region with a falling trend makes up 11.3% of the overall area when examining each grade, the area with a constant trend makes up 9.1% of the total area, and the area with a growing trend makes up 79.6% of the whole area. It is evident that the research area's predominant changing pattern is an upward trend. From Figure 9d, it can be seen that most regions of the Loess Plateau exhibit an improvement trend, with the main improvement areas located in the central part of the Loess Plateau. Analyzing each grade, the area with significant improvement is 268,087.75 km$^2$, accounting for 42.2% of the total area; the area with slight improvement is 236,856.25 km$^2$, accounting for 37.3% of the total area; the stable area is 57,810 km$^2$, accounting for 9.1% of the total area; the slightly degraded area is 58,486 km$^2$, accounting for 9.2% of the total area; and the severely degraded area is 13,463 km$^2$, accounting for 2.1% of the total area. The dominant trends are significant improvement and slight improvement, indicating an improvement in the environmental quality of the Loess Plateau, which is consistent with the previous analysis results. The degraded areas are mainly located in the Inner Mongolia Autonomous Region, the Ningxia Hui Autonomous Region, and the northern and northwestern regions of Qinghai Province. For Shaanxi, Shanxi, and Henan, the degraded areas are mostly located in urban areas, such as Xi'an, Xianyang, Taiyuan, Luoyang, and other urban areas, which is consistent with previous research findings [74,75].

Both indices indicate poor environmental quality in urban areas and desertified regions, while the central and southern regions of the Loess Plateau with high forest coverage exhibit better environmental quality. In comparison to the RSEI, the SRSEI detects a larger area of degraded regions. Specifically, the SRSEI shows 27,320.25 km$^2$ more area in severely degraded and slightly degraded regions than the RSEI, 26,858.5 km$^2$ more area in stable regions, and 54,178.75 km$^2$ less area in slightly improved and significantly improved regions. The SRSEI identifies more degraded regions than the RSEI, particularly in urban clusters, the northwestern grassland, and northern desert regions. This may be attributed to vegetation being affected by drought stress, resulting in decreased biological activity and photosynthetic capacity, which leads to a decrease in fluorescence values.

## 4. Discussion

### 4.1. Advantages of SRSEI

The composition of an ecosystem is complex, and its environmental quality is influenced by multiple components. Although using a single index to monitor certain components of the ecological e nvironment system for reflecting changes in ecological environmental quality has some value, the comprehensive effects of multiple factors acting together cannot be fully revealed [76–78]. The SRSEI addresses the problem of single and

biased evaluation indicators for the environmental quality of ecosystems by constructing indices that reflect the greenness, warmth, humidity, and dryness of the ecological environment. From Figure 2, it can be observed that the contribution rates of the first principal component (PC1) for each SRSEI exceed 67%, with the largest contribution rate being 87%. This indicates that most of the information is concentrated in the PC1 band. Therefore, the SRSEI constructed based on PC1 can better reflect the comprehensive effects of various factors on the ecological environmental quality in the study area compared to using a single index. From Figure 4, it can be seen that there is a significant and highly significant correlation between the SRSEI and the various indices, demonstrating the feasibility of using the SRSEI for regional ecological environmental quality assessment. From Table 3 it can be seen that there is a significant correlation between the SRSEI and RSEI from 2001 to 2018 and the cumulative precipitation from May to September, and the correlation coefficient between the SRSEI and cumulative precipitation is larger than that between the RSEI and cumulative precipitation. This indicates that the SRSEI is more affected by precipitation information and indirectly reflects the aridity conditions in the study area. Since fluorescence data can reflect vegetation photosynthetic activity and respond earlier to mild drought stress, incorporating fluorescence data to construct the SRSEI for environmental monitoring in the study area is effective. Whether it is the RSEI or SRSEI, the factor data used can be obtained through remote sensing inversion, facilitating large-scale, efficient, and short-period observations and spatiotemporal change analysis in the study area, as well as visualization, modeling, and prediction [79]. However, compared to the complex composition of ecosystems, the four indices used are not comprehensive enough. Therefore, it is necessary to identify more and more suitable indices to be incorporated into the construction of the SRSEI to make it more complete and objective. Although the SRSEI cannot fully reflect the ecological environmental quality of the study area, it relies entirely on remote sensing data, which are relatively easy to obtain.

### 4.2. Advantages of Utilizing the Google Earth Engine (GEE) Platform

The research area's varied remote sensing data must first be downloaded locally before being preprocessed and analyzed to create the typical RSEI and SRSEI. Instead of requiring picture downloads, the Google Earth Engine (GEE) platform maintains land satellite observation data from several data sources globally from 1984 to the present. This allows for real-time data updates, and the cloud platform has many preprocessed product data, eliminating the need for researchers to preprocess remote sensing data [80]. Furthermore, the GEE platform provides powerful online computing capabilities, allowing researchers to perform data analysis and display results on the platform's powerful servers. GEE can collect cloud-free remote sensing images for any time in the study area using cloud detection methods, ensuring high data quality and more objective and realistic monitoring results. From the results of this study, the principal component analysis results of various indices computed using the GEE platform are consistent with the actual situation. The greenness and humidity indices are positively correlated with the RSEI and SRSEI, while the warmth and dryness indices are negatively correlated with the RSEI and SRSEI (Figure 3). This indicates that utilizing GEE for the construction of the SRSEI and RSEI in the Loess Plateau region can reflect the ecological environmental quality of the study area, providing a solid foundation and data guarantee for a scientifically accurate assessment and analysis of the spatiotemporal evolution patterns of the ecological environment in the Loess Plateau. The GEE platform offers a variety of built-in remote sensing image processing codes, such as cloud detection, principal component analysis, remote sensing data band synthesis, and data clipping, as well as built-in modeling and analysis codes, such as linear regression, ridge regression, etc. [81]. These codes and functions enable researchers to avoid becoming bogged down in massive data processing, especially for global- and intercontinental-scale remote sensing analysis. Using these built-in codes, researchers can quickly and accurately capture the changes in the SRSEI in the study area and predict the future ecological environmental quality of the study area. In this study, high-quality

remote sensing data of the study area were obtained using the GEE platform, RSEIs were computed online, and then SRSEIs were constructed using fluorescence data along with the dryness, warmth, and humidity indices, greatly improving research efficiency. Trend analysis methods can be applied to explore changes in ecological environmental quality in the Loess Plateau region and analyze their evolution patterns. Through visualization, changes in specific regions can be directly depicted and quantitatively analyzed. The development of the GEE platform, which allows for the monitoring and evaluation of regional environmental quality on a larger scale and over a longer monitoring period than traditional regional environmental quality assessments based on remote sensing data, opens up a wide range of application possibilities for the monitoring, analysis, and driving force research of ecological environmental quality in the Loess Plateau region of China.

*4.3. Investigation of the Factors That Influence Changes in Ecological Environmental Quality*

Due to human interference with the natural environment, which has gotten worse in recent years as urbanization and climate change processes have intensified, the ecological environment quality of the Loess Plateau region has deteriorated. In areas prone to soil erosion and water erosion, there have been significant improvements in environmental quality. The Loess Plateau is enormous and spans several provinces, towns, and autonomous entities. Between diverse regions, which may be categorized into mountainous areas, loess hilly areas, loess tableland areas, loess terrace tableland areas, and river valley plain areas, there are notable changes in topography, landforms, and climates. Each has its own distinctive natural habitat. The Loess Plateau is located in a semi-arid and arid climatic region, characterized by significant seasonal variations in precipitation. The ecological environment is fragile, with slow vegetation recovery and vulnerability to drought impacts [82]. The vegetation coverage in the Loess Plateau shows a distinct step-like pattern, gradually increasing from the northwest to the southeast, transitioning from grassland to forest. The northern region is characterized by desert distribution and is unsuitable for vegetation growth.

According to the research in this study, the arid regions in the northern and northwestern sections of the Loess Plateau are where the majority of the regions with extremely bad and poor environmental quality are located. These areas have low vegetation coverage, poor water retention capacity, and high evaporation rates compared to actual rainfall, resulting in an overall dryness. There are also a few areas located in urban clusters where the population density is high and urbanization is advanced, leading to the urban heat island effect (Figures 6 and 7). The main causes of these phenomena are the inherent ecological vulnerability of the Loess Plateau, susceptibility to various erosional processes, and long-term impacts of unsustainable human activities. However, the Loess Plateau has shown the fastest improvement in environmental quality in the past four decades (1980–2020) (Figure 9. This is a result of key soil and water conservation projects, such as the "Eight Shields" project, key prevention and control projects for soil erosion in the middle and upper reaches of the Yellow River, and the comprehensive management of soil erosion on sloping farmland. Since the 1980s, China has implemented the Three-North Shelterbelt Project, afforestation projects, and returning farmland to forest and grassland. The construction of the Three-North Shelterbelt Project has significantly improved the environmental conditions in the Loess Plateau, with over 300 million mu (approximately 20 million hectares) of soil erosion being effectively controlled. The forest coverage in the Loess Plateau has significantly increased, and the soil erosion modulus has greatly decreased [83]. The environmental quality has notably improved in the central, eastern, and southern regions of the Loess Plateau (Figure 6), mainly due to abundant precipitation and a moderate climate, which are favorable for vegetation growth and recovery. However, there has been a decline in the ecological environmental quality in some areas (Figure 6), such as the Guanzhong Urban Agglomeration, the Fenhe River Valley Urban Agglomeration, as well as major cities like Xining, Lanzhou, Luoyang, and Yinchuan. These areas have experienced population growth and socio-economic development due to urban expansion,

leading to significant environmental degradation caused by high levels of urbanization. The Loess Plateau's future economic growth depends on having both "gold and silver mountains" and "lucid waters and lush mountains", which are both essential elements and challenges that must be overcome [84].

The Loess Plateau region's environmental quality has increased between 2001 and 2021, according to the data above. The government's emphasis on soil and water conservation and environmental governance on the Loess Plateau is primarily responsible for this improvement. The implementation of afforestation and land restoration programs has significantly increased vegetation coverage in the Loess Plateau. While pursuing economic development, it is important to also prioritize environmental protection. Therefore, all provinces, cities, and autonomous regions in the Loess Plateau should emphasize the coordination between economic development and regional ecological conservation to achieve a positive cycle of green economic development and improved environmental quality.

## 5. Conclusions

In this study, sun-induced chlorophyll fluorescence data, which represent vegetation vitality and photosynthetic capacity, were used as a new greenness indicator. Along with warmth, dryness, and humidity, they collectively constructed a new environmental monitoring index called the SRSEI. Slope trend analysis and the Mann–Kendall trend test were employed to analyze the ecological environmental changes in the Loess Plateau region from 2001 to 2021.

For modeling and analysis of the remote-sensing-based RSEI and SRSEI, the GEE cloud platform may be used, which allows for the quick and precise collection of the spatiotemporal patterns of ecological environmental quality in the Loess Plateau region. The Loess Plateau may use this widely for environmental preservation, soil and water conservation, and high-quality regional economic growth. Future research avenues have been suggested through the examination of new environmental monitoring indices based on remote sensing data. Most sections of the Loess Plateau exhibit an upward trend in the SRSEI in terms of ecological environmental quality, with areas of major improvement and modest improvement predominating. This indicates an overall improvement in environmental quality in the region, which is attributed to comprehensive environmental governance efforts in the Loess Plateau. Continued or increased efforts are needed to protect the ecological environment in order to maintain the existing achievements in environmental governance. However, it is worth noting that the process of urbanization in the study area has an impact on ecological environmental quality. As urbanization intensifies, there is a downward trend in environmental quality. Areas with ecological degradation in the study area mainly concentrate in urbanized regions and desertified areas. In the future, it is essential to prioritize both economic development and ecological environmental protection in arid and semi-arid regions.

**Author Contributions:** Conceptualization, M.S.; methodology, M.S. and X.J.; software, F.L.; validation, M.S.; formal analysis, M.S. and F.L.; investigation, F.L.; resources, Y.H.; data curation, X.J.; writing—original draft preparation, M.S.; writing—review and editing, F.L., Y.S. and X.J.; visualization, B.L.; supervision, Y.H.; project administration, Y.H.; funding acquisition, Y.H. All authors have read and agreed to the published version of the manuscript.

**Funding:** This research was funded by the National Key Research and Development Program (grant number 2021YFD2000200) and the National Natural Science Foundation of China (grant number 42171394).

**Institutional Review Board Statement:** Not applicable.

**Informed Consent Statement:** Not applicable.

**Data Availability Statement:** Not applicable.

**Acknowledgments:** We would like to extend our sincere thanks to Jingfeng Xiao and Xing Li of the University of New Hampshire. The authors thank them for providing the GOSIF data.

**Conflicts of Interest:** The authors declare no conflict of interest.

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
