# Peer review of "Ecological Environment Quality Assessment of Arid Areas Based on Improved Remote Sensing Ecological Index—A Case Study of the Loess Plateau"

_sustainability, doi:10.3390/su151813881_

Round 1

Reviewer 1 Report

1 Introduction

(1) In the first paragraph, please provide more background on the study of the ecosystem and the reality of the study of the Loess Plateau ecosystem. It is important to note that the context should be global and not limited to the Loess Plateau or China. This is important for an international paper.

(2) Prior to the advent of RSEI, analyses of ecosystems were commonly characterized by land use change, NDVI change, and ecosystem service change, and authors were asked to add to and comment on this, and then logically derive a literature review of RSEI.

(3) In the literature review of RSEI, please add more studies by non-Chinese scholars, which have shown that the use of RSEI to study ecological and environmental issues is not only limited to China, but has also been utilized in other countries. This could provide more evidence for the use of RSEI by foreign scholars in the future.

(4) In the third paragraph of the introduction, the applicability of SIF in arid zones has been emphasized in the central point, so the authors are asked to consider whether the title of the paper can be changed from "Loess Plateau" to "Arid Zones" in order to enhance the attractiveness and influence of the article (provided that it can be successfully published). If not, then the authors should change the title of the paper to "Arid Zone" in the citation.) If not, then the authors must emphasize in the introduction that the Loess Plateau is located in a semi-arid arid region in order to emphasize the need for SIF.

(5) The sudden appearance of GEE in the last paragraph is abrupt and has no logical connection with the above. To elucidate the merits of GEE, one must stand for the shortcomings caused by the failure of previous studies to use GEE. The author is requested to add to the research deficiencies that may have been caused by the failure of previous scholars to use GEE in their studies.

2.1 study area

Figure 1 does not conform to international journal cartographic standards and the quality of all the figures throughout the text needs to be improved. Inner Mongolia in Figure 1 (a) is incorrectly written, please change it!

3 Results and discussion

(1) Please address the results section and the discussion section separately.

(2) In the discussion of the results, the author only briefly mentions that they are in line with the results of the previous studies, without discussing and explaining the other results obtained by the paper. Since the paper is a methodology-based paper, it is important to discuss in depth in the discussion how the results of the paper's research differ from the results of previous research, and in particular, it is necessary to answer why they are different. Does the new methodology proposed in the dissertation address the shortcomings that existed in the previous studies as compared to the results of the previous studies? Are the results obtained with the new methodology more in line with the actual situation in the study area? Can the new method be generalized to other arid regions in the world? What are the shortcomings of the paper and what are the prospects for future research? All of the above are key points that need to be discussed in depth by the authors, so please think carefully and make detailed additions.

(3) In section 3.5, both RSEI and SRSEI for the Loess Plateau should be analyzed for slope trends with local zoom in order to more clearly illustrate the differences between the two methods, especially the need for comparative analyses to show the advantages of SRESI.

4 Conclusion

The conclusion is not written like a conclusion, but as if the author is piling up results. In the conclusion section, please organize it logically around the question "Do the results of the paper answer the scientific question of the paper"

 Moderate editing of English language required.

Author Response

Dear reviewer,

Hello. I have made the necessary revisions to the manuscript and have uploaded the relevant files. Please review them at your convenience. Thank you.

Best regards, [Ming Shi]

Reviewer 2 Report

The article titled:  Ecological Environment Analysis of Loess Plateau Based on Modified Remote Sensing Ecological Index. The study generated remote sensing ecological indices and fluorescence remote sensing remote sensing ecological indices for the Loess Plateau region from 2001 to 2021 using Growing Season Integrated Fluorescence (GOSIF) and MODIS products. The manuscript is well written and well presented. The proposed objective is met according to the results presented. After all comments have been discussed, in order to improve the quality of the manuscript, it could be considered for publication.

General comments:

1.    All figures need quality improvement (higher resolution).

2.    Revise the English version of the manuscript

Specific Comments

1.Abstract: Do not use abbreviations in this section

2.Introduction: increase the number of references to at least 20 in this section

3. equation 2 is out of sync with the text (I suggest reducing the significant figures after the comma).

4. In table 3 unify the significant digits

5. Figure 4 is not understandable, increase the size of the letters or numbers, or separate it into several figures.

6. Figures 5 and 6 improve resolution

7. Figure 8 nothing is understood

8. Figure 9a and 9b perhaps use different colors in the legends so as not to create confusion. For example, the mustard color in Fig 9a is Rise and in Fig 9b is sligh degradation.

Success in your revisions!!!

No comments

Author Response

Dear Reviewer,

Hello. I have made the necessary revisions to the manuscript and have uploaded the relevant files. Please review them at your convenience. Thank you.

Best regards, [Ming Shi]

Reviewer 3 Report

This study aims to assess the ecological environment quality of China's ecologically fragile Loess Plateau region by utilizing a remote sensing-based ecological evaluation method called the Remote Sensing Ecology Index (RSEI). This index incorporates Sun-induced chlorophyll fluorescence (SIF) data, a new vegetation remote sensing technology, to overcome the limitations of traditional vegetation indices. By employing Google Earth Engine (GEE) for geospatial analysis, the study aims to analyze spatiotemporal changes in ecological environment quality from 2001 to 2021. The goal is to offer a theoretical foundation and methodological framework for monitoring, evaluating, and safeguarding the ecological environment quality in the Loess Plateau region. The study builds upon previous research utilizing remote sensing ecological indices, adapting them to include SIF data, and using GEE for comprehensive analysis.

I have the following comments and suggestions for including information and improvements:

Abstract

The authors must include additional information about the methods, and central result values. It is imperative that the authors directly and objectively present the study's objectives more explicitly. Additionally, the authors must adhere to the guidelines; the abstract should be at most 200 words, while the current version has 390 words. Please adjust it according to the guidelines (https://www.mdpi.com/journal/sustainability/instructions). The current abstract version should reflect this aspect, resulting in a more concise and robust abstract.

Introduction

The authors should illustrate more about state of the art in analysis and methods to estimate RSEI and SIF in other regions worldwide.

A more explicit justification must be included regarding the Importance of testing this approach in different counties or locations with similar characteristics. This information is essential to give a more general aspect to the study, helping it not appear that the results are only of local Importance.

Experimental Materials and Methods

Please include a flowchart of the experiment/analysis. This inclusion is necessary to enhance the presentation quality of our methods and to provide better clarity for readers in understanding the results.

More details must be included in the sections:

1. Please describe in more detail the sections: Remote sensing ecological monitoring indicators, Greenness indicators, Humidity index, Dryness index, Heat index, SIF index, Construction of SRSEI. Please describe in more detail the sections: Remote Sensing Ecological Monitoring Indicators, Greenness Indicators, Moisture Index, Dryness Index, Heat Index, SIF Index, SRSEI Construction. What are the criteria for choosing the indices? What are the assumptions of the methods used? What is the criterion for choosing the sensor systems used? How were the index results validated? What is the criterion for dividing categories after normalization?

Results/ Discussion

1. All results derived from the Principal component analysis of SRSEI indicators, RSEI versus SRSEI in the Loess Plateau, The spatial distribution of RSEI and SRSEI in the Loess Plateau region, Comparison of RSEI and SRSEI levels among provinces in the Loess Plateau region, Trends of SRSEI in the Loess Plateau must be detailed (More details are necessary in all sections).

2. Please replace figures 2,3,4,5,6,7, and 8 with better ones (Increase the resolution quality and font size of the graph axis and other values).

Citations that support the results found in this section must be included. It is also imperative that the authors have a topic that compares the results found from the methods and analyses proposed in the article with the literature results.

Please include a paragraph with possible limitations of the method exposed in your study and the extrapolation of these methods to other regions of worldwide.

Conclusions

In this section, information should be added about future advances in this field of research based on the results found in this article.

Author Response

(The authors gave the same response as above.)

Reviewer 4 Report

Review for “Ecological Environment Analysis of Loess Plateau Based on Modified Remote Sensing Ecological Index”

A new ecological environment index called SRSEI was constructed by using SIF data to replace the vegetation index in RSEI. SRSEI showed a higher correlation with rainfall data and indicated an improvement in the ecological environment quality of the Loess Plateau region. The areas with significantly improved and slightly improved ecological environment quality accounted for the largest proportion. The quality of the ecological environment in the research region may be assessed accurately using the newly developed ecological environment index.

The paper is well-written and the topic looks intriguing. I recommend a "Major" change before applying to Sustainability. I wrote:

1.     I checked similarity and it was 30%, where 10% only from this manuscript (https://doi.org/10.3390/land11081306)

2.     The introduction section needs to be organized in subtitles and more newly references should be added to the introduction and further discussion is needed.

3.     Overall figures need to be remade as almost all figures are not readable. Please increase the text size in figure 1, 2, 4, 5, 6, 8, 8

4.     Add a flowchart to describe the method used in the manuscript.

5.     Table 1: make same spatial resolution

6.     Please double-check the citations; I was unable to access some of them since the corresponding DOIs were either missing or incorrectly placed. Inspect them thoroughly.

7.     Although the manuscript is generally good written, a language check by a professional native speaker or an editing agency is needed to fix some syntax, style, and phrasing problems.

I look forward to seeing a better version of the manuscript.

Although the manuscript is generally good written, a language check by a professional native speaker or an editing agency is needed to fix some syntax, style, and phrasing problems.

Author Response

(The authors gave the same response as above.)

Round 2

Reviewer 1 Report

It is clear that the author has carefully revised the manuscript and has done a good job of fixing the problems that existed before, and the quality of the manuscript has been greatly improved. Congratulations to the authors!

Reviewer 2 Report

Congratulations!!!

Reviewer 3 Report

We wanted to express our gratitude to the authors for their hard work in addressing all of the comments and making the necessary modifications to the paper. We are pleased to inform the authors that the current version of the paper with the ID sustainability-2591986 is now ready for acceptance and publication in the Sustainability journal. 

Reviewer 4 Report

Accepted